# CHEMICAL-REACTION-AWARE MOLECULE REPRESENTATION LEARNING

**Hongwei Wang[1], Weijiang Li[1], Xiaomeng Jin[1], Kyunghyun Cho[2,3], Heng Ji[1], Jiawei Han[1], Martin D. Burke[1]**
[1]University of Illinois Urbana-Champaign, [2]New York University, [3]Genentech
{hongweiw, wl13, xjin17, hengji, hanj, mdburke}@illinois.edu, kyunghyun.cho@nyu.edu

## ABSTRACT

Molecule representation learning (MRL) methods aim to embed molecules into a real vector space. However, existing SMILES-based (Simplified Molecular-Input Line-Entry System) or GNN-based (Graph Neural Networks) MRL methods either take SMILES strings as input that have difficulty in encoding molecule structure information, or over-emphasize the importance of GNN architectures but neglect their generalization ability. Here we propose using chemical reactions to assist learning molecule representation. The key idea of our approach is to preserve the equivalence of molecules with respect to chemical reactions in the embedding space, i.e., forcing the sum of reactant embeddings and the sum of product embeddings to be equal for each chemical equation. This constraint is proven effective to 1) keep the embedding space well-organized and 2) improve the generalization ability of molecule embeddings. Moreover, our model can use any GNN as the molecule encoder and is thus agnostic to GNN architectures. Experimental results demonstrate that our method achieves state-of-the-art performance in a variety of downstream tasks, e.g., reaction product prediction, molecule property prediction, reaction classification, and graph-edit-distance prediction. The code is available at https://github.com/hwwang55/MolR.

## 1 INTRODUCTION

How to represent molecules is a fundamental and crucial problem in chemistry. Chemists usually use IUPAC nomenclature, molecular formula, structural formula, skeletal formula, etc., to represent molecules in chemistry literature.[1] However, such representations are initially designed for human readers rather than computers. To facilitate machine learning algorithms understanding and making use of molecules, *molecule representation learning* (MRL) is proposed to map molecules into a low-dimensional real space and represent them as dense vectors. The learned vectors (a.k.a. embeddings) of molecules can benefit a wide range of downstream tasks, such as chemical reaction prediction (Jin et al., 2017; Segler & Waller, 2017), molecule property prediction (Zhang et al., 2021), molecule generation (Mahmood et al., 2021), drug discovery (Rathi et al., 2019), retrosynthesis planning (Segler et al., 2018), chemical text mining (Krallinger et al., 2017), and chemical knowledge graph modeling (Bean et al., 2017).

Researchers have proposed a great many MRL methods. A large portion of them, including Mol-BERT (Fabian et al., 2020), ChemBERTa (Chithrananda et al., 2020), SMILES-Transformer (Honda et al., 2019), SMILES-BERT (Wang et al., 2019), Molecule-Transformer (Shin et al., 2019), and SA-BiLSTM (Zheng et al., 2019b), take *SMILES*[2] strings as input and utilize natural language models, for example, Transformers (Vaswani et al., 2017) or BERT (Devlin et al., 2018), as their base model. Despite the great power of such language models, they have difficulty dealing with SMILES input,

---

[1]For example, for glycerol, its IUPAC name, molecular formula, structural formula, and skeletal formula are propane-1,2,3-triol, $C_3H_8O_3$, $\overset{OH}{\underset{|}{CH_2}} - \overset{OH}{\underset{|}{CH}} - \overset{OH}{\underset{|}{CH_2}}$, and $HO \diagdown \diagup \overset{OH}{} OH$, respectively.

[2]The Simplified Molecular-Input Line-Entry System (SMILES) is a specification in the form of a line notation for describing the structure of chemical species using short ASCII strings. For example, the SMILES string for glycerol is "OCC(O)CO".

because SMILES is 1D linearization of molecular structure, which makes it hard for language models to learn the original structural information of molecules simply based on "slender" strings (see Section 4 for more discussion). Another line of MRL methods, instead, use *graph neural networks* (GNNs) (Kipf & Welling, 2017) to process molecular graphs (Merkwirth & Lengauer, 2005; Jin et al., 2017; Gilmer et al., 2017; Ishida et al., 2021). Though GNN-based methods are theoretically superior to SMILES-based methods in learning molecule structure, they are limited to designing fresh and delicate GNN architectures while ignoring the essence of MRL, which is *generalization ability*. Actually, we will show later that, there is no specific GNN that performs universally best in all downstream tasks of MRL, which inspires us to explore beyond GNN architectures.

To address the limitations of existing work, in this paper, we propose using *chemical reactions* to assist learning molecule representations and improving their generalization ability. A chemical reaction is usually represented as a chemical equation in the form of symbols and formulae, wherein the reactant entities are given on the left-hand side and the product entities on the right-hand side. For example, the chemical equation of Fischer esterification of acetic acid and ethanol can be written as $CH_3COOH + C_2H_5OH \rightarrow CH_3COOC_2H_5 + H_2O$. A chemical reaction usually indicates a particular relation of *equivalence* between its reactants and products (e.g., in terms of conservation of mass and conservation of charge), and our idea is to preserve this equivalence in the molecule embedding space. Specifically, given the chemical reaction of Fischer esterification above, we hope that the equation $h_{CH_3COOH} + h_{C_2H_5OH} = h_{CH_3COOC_2H_5} + h_{H_2O}$ also holds, where $h_{(\cdot)}$ represents molecule embedding function. This simple constraint endows molecule embeddings with very nice properties: (1) Molecule embeddings are composable with respect to chemical reactions, which make the embedding space well-organized (see Proposition 1); (2) More importantly, we will show later that, when the molecule encoder is a GNN with summation as the readout function, our model can automatically and implicitly learn *reaction templates* that summarize a group of chemical reactions within the same category (see Proposition 2). The ability of learning reaction templates is the key to improving the generalization ability of molecule representation, since the model can easily generalize its learned knowledge to a molecule that is unseen but belongs to the same category as or shares the similar structure with a known molecule.

We show that the molecule embeddings learned by our proposed model, namely **MolR** (chemical-**r**eaction-aware **mol**ecule embeddings), is able to benefit a variety of downstream tasks, which makes it significantly distinct from all existing methods that are designed for only one downstream task. For example, MolR achieves 17.4% absolute Hit@1 gain in reaction product prediction, 2.3% absolute AUC gain on BBBP dataset in molecule property prediction, and 18.5% relative RMSE gain in graph-edit-distance prediction, respectively, over the best baseline method. We also visualize the learned molecule embeddings and show that they are able to encode reaction templates as well as several key molecule attributes, e.g., molecule size and the number of smallest rings.

## 2 THE PROPOSED METHOD

### 2.1 STRUCTURAL MOLECULE ENCODER

A molecular graph is represented as $G = (V, E)$, where $V = \{a_1, \cdots\}$ is the set of non-hydrogen atoms and $E = \{b_1, \cdots\}$ is the set of bonds. Each atom $a_i$ has an initial feature vector $x_i$ encoding its properties. In this work, we use four types of atom properties: *element type*, *charge*, *whether the atom is an aromatic ring*, and *the count of attached hydrogen atom(s)*. Each type of atom properties is represented as a one-hot vector, and we add an additional "unknown" entry for each one-hot vector to handle unknown values during inference. The four one-hot vectors are concatenated as the initial atom feature. In addition, each bond $b_i$ has a bond type (e.g., single, double). Since the bond type can usually be inferred by the features of its two associated atoms and does not consistently improve the model performance according to our experiments, we do not explicitly take bond type as input.

To learn structural representation of molecules, we choose GNNs, which utilize molecule structure and atom features to learn a representation vector for each atom and the entire molecule, as our base model. Typical GNNs follow a neighborhood aggregation strategy, which iteratively updates the representation of an atom by aggregating representations of its neighbors and itself. Formally, the $k$-th layer of a GNN is:

$$h_i^k = \text{AGGREGATE}\left(\left\{h_j^{k-1}\right\}_{j \in \mathcal{N}(i) \cup \{i\}}\right), \ k = 1, \cdots, K, \tag{1}$$

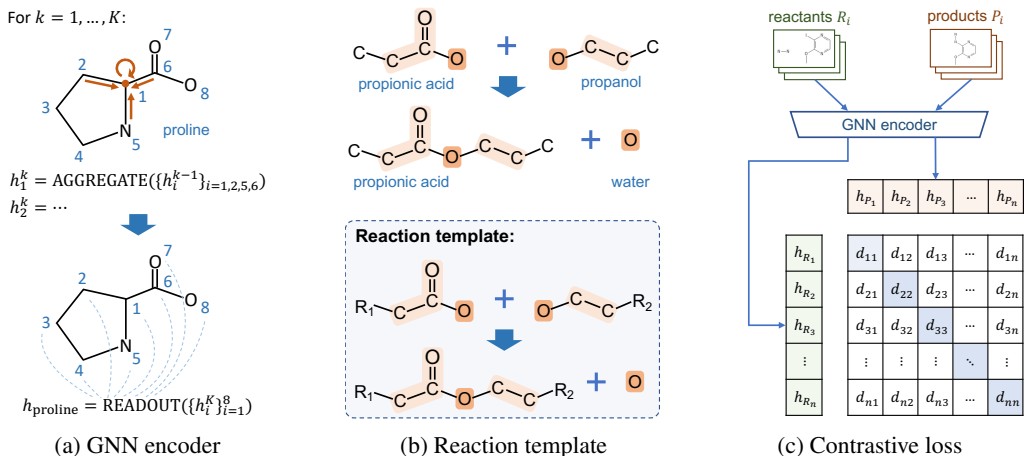

(a) GNN encoder        (b) Reaction template        (c) Contrastive loss

Figure 1: (a) Illustration of a GNN encoder processing a proline molecule. Hydrogen is omitted. (b) Illustration of Fischer esterification of propionic acid and propanol, and the corresponding reaction template learned by our model. The reaction center is colored in orange, and atoms whose distance from the reaction center is 1 or 2 are colored in light orange. (c) Illustration of the contrastive loss for a minibatch of chemical reactions. $d_{ij}$ is Euclidean distance between embedding $h_{R_i}$ and $h_{P_j}$.

where $h_i^k$ is atom $a_i$'s representation vector at the $k$-th layer ($h_i^0$ is initialized as $a_i$'s initial feature $x_i$), $\mathcal{N}(i)$ is the set of atoms directly connected to $a_i$, and $K$ is the number of GNN layers. The choice of AGGREGATE function is essential to designing GNNs, and a number of GNN architectures have been proposed. See Appendix A for a detailed introduction on GNN architectures.

Finally, a readout function is used to aggregate all node representations output by the last GNN layer to obtain the entire molecule's representation $h_G$:

$$h_G = \text{READOUT}\big(\{h_i^K\}_{a_i \in V}\big). \tag{2}$$

The READOUT function can be a simple permutation invariant function such as summation and mean, or a more sophisticated graph-level pooling algorithm (Ying et al., 2018; Zhang et al., 2018). An illustrative example of GNN encoder is shown in Figure 1a.

## 2.2 PRESERVING CHEMICAL REACTION EQUIVALENCE

A chemical reaction defines a particular relation "$\rightarrow$" between reactant set $R = \{r_1, r_2, \cdots\}$ and product set $P = \{p_1, p_2, \cdots\}$:

$$r_1 + r_2 + \cdots \rightarrow p_1 + p_2 + \cdots. \tag{3}$$

A chemical reaction usually represents a closed system where several physical quantities of the system retain constant before and after the reaction, such as mass, energy, charge, etc. Therefore, it describes a certain kind of equivalence between its reactants and products in the chemical reaction space. Our key idea is to preserve such equivalence in the molecule embedding space:

$$\sum_{r \in R} h_r = \sum_{p \in P} h_p. \tag{4}$$

The above simple constraint is crucial to improving the quality of molecule embeddings. We first show, through the following proposition, that the chemical reaction relation "$\rightarrow$" is an *equivalence relation* under the constraint of Eq. (4):

**Proposition 1** *Let $M$ be the set of molecules, $R \subseteq M$ and $P \subseteq M$ be the reactant set and product set of a chemical reaction, respectively. If $R \rightarrow P \Leftrightarrow \sum_{r \in R} h_r = \sum_{p \in P} h_p$ for all chemical reactions, then "$\rightarrow$" is an equivalence relation on $2^M$ that satisfies the following three properties: (1) Reflexivity: $A \rightarrow A$, for all $A \in 2^M$; (2) Symmetry: $A \rightarrow B \Leftrightarrow B \rightarrow A$, for all $A, B \in 2^M$; (3) Transitivity: If $A \rightarrow B$ and $B \rightarrow C$, then $A \rightarrow C$, for all $A, B, C \in 2^M$.*

The proof of Proposition 1 is in Appendix B. One important corollary of Proposition 1 is that, the set of all subsets of $M$, i.e. $2^M$, is naturally split into *equivalence classes* based on the equivalence

relation "$\rightarrow$". For all molecule sets within one equivalent class, the sum of embeddings of all molecules they consist of should be equal. For example, in organic synthesis, a target compound $t$ may be made from three different sets of starting materials $A$, $B$, and $C$. Then the sets $A$, $B$, $C$ as well as $\{t\}$ belong to one equivalence class, and we have $\sum_{m \in A} h_m = \sum_{m \in B} h_m = \sum_{m \in C} h_m = h_t$. Note that the starting materials are usually small and basic molecules that frequently appear in a number of synthesis routes. Therefore, Eq. (4) forms a system of linear equations, wherein the chemical reaction equivalence imposes strong constraint on the embeddings of base molecules. As a result, the feasible solutions of molecule embeddings will be more robust, and the whole embedding space will be more organized. See the visualized result on molecule embedding space in Section 3.5 for more details.

We can further show that, the constraint in Eq. (4) is also able to improve the generalization ability of molecule embeddings. To see this, we first define *reaction center* for a chemical reaction. The reaction center of $R \rightarrow P$ is defined as a subgraph of $R$ consisting of atoms whose bonds have changed after reaction. For example, for the reaction in the upper part of Figure 1b, its reaction center is the two oxygen atoms marked in dark orange, since they are the only two atoms whose bonds have changed. Given the concept of reaction center, we have the following proposition:

**Proposition 2** *Let $R \rightarrow P$ be a chemical reaction where $R$ is the reactant set and $P$ is the product set, and $C$ be its reaction center. Suppose that we use the GNN (whose number of layers is $K$) shown in Eqs. (1) and (2) as the molecule encoder, and set the READOUT function in Eq. (2) as summation. Then for an arbitrary atom $a$ in one of the reactant whose final representation is $h_a^K$, the residual term $\sum_{r \in R} h_r - \sum_{p \in P} h_p$ is a function of $h_a^K$ if and only if the distance between atom $a$ and reaction center $C$ is less than $K$.*

The proof of Proposition 2 is in Appendix C. Proposition 2 indicates that the residual between reactant embedding and product embedding will fully and only depend on atoms that are less than $K$ hops away from the reaction center. For example, as shown in Figure 1b, suppose that we use a 3-layer GNN to process Fischer esterification of propionic acid and propanol, then the residual between reactant embedding and product embedding will totally depend on the reaction center (colored in orange) as well as atoms whose distance from the reaction center is 1 or 2 (colored in light orange). This implies that, if the GNN encoder has been well-optimized on this chemical equation and outputs perfect embeddings, i.e., $h_{CH_3CH_2COOH} + h_{CH_3CH_2CH_2OH} = h_{CH_3CH_2COOCH_2CH_2CH_3} + h_{H_2O}$, then the equation $h_{R_1-CH_2COOH} + h_{R_2-CH_2CH_2OH} = h_{R_1-CH_2COOCH_2CH_2-R_2} + h_{H_2O}$ will also hold for any functional group $R_1$ and $R_2$, since the residual between the two sides of the equation does not depend on $R_1$ or $R_2$ that are more than 2 hops away from the reaction center. The induced general chemical reaction $R_1$-$CH_2COOH + R_2$-$CH_2CH_2OH \rightarrow R_1$-$CH_2COOCH_2CH_2$-$R_2 + H_2O$ is called a *reaction template*, which abstracts a group of chemical reactions within the same category. The learned reaction templates are essential to improving the generalization ability of our model, as the model can easily apply this knowledge to reactions that are unseen in training data but comply with a known reaction template (e.g., acetic acid plus propanol, butyric acid plus butanol). We will further show in Section 3.5 how reaction templates are encoded in molecule embeddings.

**Remarks**. Below are some of our remarks to provide further understanding on the proposed model:

First, compared with (Jin et al., 2017) that also learns reaction templates, our model does not need a complicated network to calculate attention scores, nor requires additional atom mapping information between reactants and products as input. Moreover, our model is theoretically able to learn a reaction template based on even only one reaction instance, which makes it particularly useful in few-shot learning (Wang et al., 2020) scenario. See Section 3.1 for the experimental result.

Second, organic reactions are usually imbalanced and omit small and inorganic molecules to highlight the product of interest (e.g., $H_2O$ is usually omitted from Fischer esterification). Nonetheless, our model can still learn meaningful reaction templates as long as chemical reactions are written in a consistent manner (e.g., $H_2O$ is omitted for all Fischer esterification).

Third, the number of GNN layers $K$ can greatly impact the learned reaction templates according to Proposition 2: A small $K$ may not be enough to represent a meaningful reaction template (e.g., in Fischer esterification, the necessary carbonyl group "$C=O$" in carboxylic acid will not appear in the reaction template if $K < 3$), while a large $K$ may include unnecessary atoms for a reaction template and thus reduces its coverage (e.g., the Fischer esterification of formic acid HCOOH and methanol

$CH_3OH$ is not covered by the reaction template shown in Figure 1b). The empirical impact of $K$ is shown in Appendix D.

## 2.3 TRAINING THE MODEL

According to Eq. (4), a straightforward loss function for the proposed method is therefore $L = \frac{1}{|\mathcal{D}|} \sum_{(R \to P) \in \mathcal{D}} \left\| \sum_{r \in R} h_r - \sum_{p \in P} h_p \right\|_2$, where $R \to P$ represents a chemical reaction in the training data $\mathcal{D}$. However, simply minimizing the above loss does not work, since the model will degenerate by outputting all-zero embeddings for all molecules. Common solutions to this problem are introducing negative sampling strategy (Goldberg & Levy, 2014) or contrastive learning (Jaiswal et al., 2021). Here we use a minibatch-based contrastive learning framework similar to (Radford et al., 2021) since it is more time- and memory-efficient.

For a minibatch of data $\mathcal{B} = \{R_1 \to P_1, R_2 \to P_2, \cdots\} \subseteq \mathcal{D}$, we first use the GNN encoder to process all reactants $R_i$ and products $P_i$ in this minibatch and get their embeddings. The matched reactant-product pairs $(R_i, P_i)$ are treated as positive pairs, whose embedding discrepancy will be minimized, while the unmatched reactant-product pairs $(R_i, P_j)$ $(i \neq j)$ are treated as negative pairs, whose embedding discrepancy will be maximized. To avoid the total loss being dominant by negative pairs, similar to (Bordes et al., 2013), we use a margin-based loss as follows (see Figure 1c for an illustrative example):

$$L_{\mathcal{B}} = \frac{1}{|\mathcal{B}|} \sum_i \left\| \sum_{r \in R_i} h_r - \sum_{p \in P_i} h_p \right\|_2 + \frac{1}{|\mathcal{B}|(|\mathcal{B}| - 1)} \sum_{i \neq j} \max \left( \gamma - \left\| \sum_{r \in R_i} h_r - \sum_{p \in P_j} h_p \right\|_2, 0 \right), \quad (5)$$

where $\gamma > 0$ is a margin hyperparameter. The whole model can thus be trained by minimizing the above loss using gradient-based optimization methods such as stochastic gradient descent (SGD).

Eq. (5) can be seen as a special negative sampling strategy, which uses all unmatched products in a minibatch as negative samples for a given reactant. Note, however, that it has two advantages over traditional negative sampling (Mikolov et al., 2013): (1) No extra memory is required to store negative samples; (2) Negative samples are automatically updated at the beginning of a new epoch since training instances are shuffled, which saves time of manually re-sampling negative samples.

## 3 EXPERIMENTS

### 3.1 REACTION PRODUCT PREDICTION

**Dataset**. We use reactions from USPTO granted patents collected by Lowe (2012) as the dataset, which is further cleaned by Zheng et al. (2019a). The dataset contains 478,612 chemical reactions, and is split into training, validation, and test set of 408,673, 29,973, and 39,966 reactions, respectively, so we refer to this dataset as *USPTO-479k*. Each reaction instance in USPTO-479k contains SMILES strings of up to five reactant(s) and exactly one product.

**Evaluation Protocol**. We formulate the task of reaction product prediction as a ranking problem. In the inference stage, given the reactant set $R$ of a chemical reaction, we treat all products in the test set as a candidate pool $C$ (which contains 39,459 unique candidates), and rank all candidates based on the L2 distance between reactant embeddings $h_R$ and candidate product embeddings $\{h_c\}_{c \in C}$, i.e., $d(R, c) = \|h_R - h_c\|_2$. Then the ranking of the ground-truth product can be used to calculate *MRR* (mean reciprocal rank), *MR* (mean rank), and *Hit@1, 3, 5, 10* (hit ratio with cut-off values of 1, 3, 5, and 10). Each experiment is repeated 3 times, and we report the result of mean and standard deviation on the test set when MRR on the validation set is maximized.

**Baselines**. We use *Mol2vec* (Jaeger et al., 2018) and *MolBERT* (Fabian et al., 2020) as baselines. For each baseline, we use the released pretrained model to output embeddings of reactants $h_R$ and candidate products $\{h_c\}_{c \in C}$, then rank all candidates based on their dot product: $d(R, c) = -h_R^\top h_c$. Since the pretrained models are not finetuned on USPTO-479k, we propose two finetuning strategies: *Mol2vec-FT1* and *MolBERT-FT1*, which freeze their model parameters but train a diagonal matrix $K$ to rank candidates: $d(R, c) = -h_R^\top K h_c$; *MolBERT-FT2*, which finetunes its model parameters by minimizing the contrastive loss function as shown in Eq. (5) on USPTO-497k. Note that Mol2vec is not an end-to-end model and thus cannot be finetuned using this strategy.

| Metrics | MRR | MR | Hit@1 | Hit@3 | Hit@5 | Hit@10 |
|---|---|---|---|---|---|---|
| Mol2vec | 0.681 | 483.7 | 0.614 | 0.725 | 0.759 | 0.798 |
| Mol2vec-FT1 | $0.688 \pm 0.000$ | $\underline{417.6} \pm 0.1$ | $0.620 \pm 0.000$ | $0.734 \pm 0.000$ | $0.767 \pm 0.000$ | $0.806 \pm 0.000$ |
| MolBERT | 0.708 | 460.7 | 0.623 | 0.768 | 0.811 | 0.858 |
| MolBERT-FT1 | $0.731 \pm 0.000$ | $457.9 \pm 0.0$ | $0.649 \pm 0.000$ | $0.790 \pm 0.000$ | $0.831 \pm 0.000$ | $0.873 \pm 0.000$ |
| MolBERT-FT2 | $\underline{0.776} \pm 0.000$ | $459.6 \pm 0.2$ | $\underline{0.708} \pm 0.000$ | $\underline{0.827} \pm 0.000$ | $\underline{0.859} \pm 0.000$ | $\underline{0.891} \pm 0.000$ |
| MolR-GCN | $0.905 \pm 0.001$ | $34.5 \pm 2.4$ | $0.867 \pm 0.001$ | $0.938 \pm 0.001$ | $0.950 \pm 0.001$ | $0.961 \pm 0.002$ |
| MolR-GAT | $0.903 \pm 0.002$ | $35.3 \pm 2.8$ | $0.864 \pm 0.002$ | $0.935 \pm 0.003$ | $0.948 \pm 0.003$ | $0.961 \pm 0.003$ |
| MolR-SAGE | $0.903 \pm 0.004$ | $53.0 \pm 4.6$ | $0.865 \pm 0.005$ | $0.935 \pm 0.004$ | $0.948 \pm 0.004$ | $0.961 \pm 0.002$ |
| MolR-TAG | $\mathbf{0.918} \pm 0.000$ | $\mathbf{27.4} \pm 0.4$ | $\mathbf{0.882} \pm 0.000$ | $\mathbf{0.949} \pm 0.001$ | $\mathbf{0.960} \pm 0.001$ | $\mathbf{0.970} \pm 0.000$ |
| MolR-TAG (1% training data) | $0.904 \pm 0.002$ | $33.0 \pm 3.7$ | $0.865 \pm 0.003$ | $0.937 \pm 0.003$ | $0.951 \pm 0.002$ | $0.963 \pm 0.002$ |

Table 1: Result of reaction product prediction on USPTO-479k dataset. The best results are highlighted in bold and the best results of baselines are highlighted with underline.

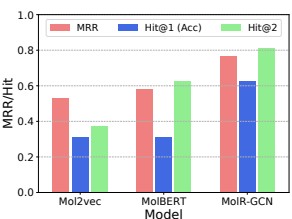

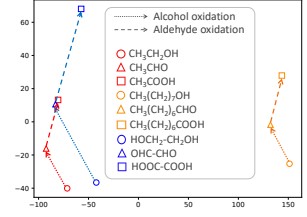

| Feature mode | Concat | Subtract |
|---|---|---|
| Mol2vec | $1.140 \pm 0.041$ | $0.995 \pm 0.034$ |
| MolBERT | $1.127 \pm 0.042$ | $0.937 \pm 0.029$ |
| MolR-GCN | $0.976 \pm 0.026$ | $0.922 \pm 0.019$ |
| MolR-GAT | $1.007 \pm 0.021$ | $0.943 \pm 0.016$ |
| MolR-SAGE | $\mathbf{0.918} \pm 0.028$ | $\mathbf{0.817} \pm 0.013$ |
| MolR-TAG | $0.960 \pm 0.027$ | $0.911 \pm 0.027$ |

Figure 2: Result of answering real multi-choice questions on product prediction.

Figure 3: Visualized reactions of alcohol oxidation and aldehyde oxidation.

Table 2: Result of RMSE in GED prediction. The best results are highlighted in bold.

**Hyperparameters Setting**. We use the following four GNNs as our molecule encoder: *GCN* (Kipf & Welling, 2017), *GAT* (Veličković et al., 2018), *SAGE* (Hamilton et al., 2017), and *TAG* (Du et al., 2017), of which the details are introduced in Appendix A. The number of layers for all GNNs is 2, the output dimension of all layers is 1,024, and the READOUT function is sum. The margin $\gamma$ is set to 4. We train the model for 20 epochs with a batch size of 4,096, using Adam (Kingma & Ba, 2015) optimizer with a learning rate of $10^{-4}$. The result of hyperparameter sensitivity and ablation study are in Appendix D and E, respectively.

**Result**. The result is reported in Table 1. All the four variants of MolR significantly outperform baseline methods. For example, MolR-TAG achieves 14.2% absolute MRR gain and 17.4% absolute Hit@1 gain over the best baseline MolBERT-FT2. In addition, we train MolR-TAG on only the first 4,096 (1%) reaction instances in the training set (with a learning rate of $10^{-3}$ and 60 epochs while other hyperparameters remain the same), and the performance degrades quite slightly. This justifies our claim in Remark 1 in Section 2.2 that MolR works well in few-shot learning scenario.

To test the performance of MolR in realistic scenarios, we collect 16 multi-choice questions on product prediction from online resources of GRE Chemistry Test Practice Book etc. Each question provides the reactants of a reaction and asks to select the correct product out of 4 or 5 choices. These multi-choice questions are quite difficult even for chemists, since the candidates are usually very similar to each other (see Appendix F for details). The result is shown in Figure 2, which indicates that MolR surpasses baselines by a large margin. Specifically, the Hit@1 (accuracy) of MolR-GCN is 0.625, which is twice as much as Mol2vec and MolBERT.

Moreover, we also conduct a case study on the first 20 reactions in the test set of USPTO-479k. Please refer to Appendix G for details.

## 3.2 MOLECULE PROPERTY PREDICTION

**Datasets**. We evaluate MolR on five datasets: *BBBP*, *HIV*, *BACE*, *Tox21*, and *ClinTox*, proposed by Wu et al. (2018). Each dataset contains thousands of molecule SMILES as well as binary labels indicating the property of interest (e.g., Tox21 measures the toxicity of compounds).

**Baselines**. We compare our method with the following baselines: *SMILES-Transformers* (Honda et al., 2019), *ECFP4* (Rogers & Hahn, 2010), *GraphConv* (Duvenaud et al., 2015), *Weave* (Kearnes et al., 2016), *ChemBERTa* (Chithrananda et al., 2020), *D-MPNN* (Yang et al., 2019), *CDDD* (Winter

| Datasets | BBBP | HIV | BACE | Tox21 | ClinTox |
|---|---|---|---|---|---|
| SMILES-Transformers | 0.704 | 0.729 | 0.701 | 0.802 | **0.954** |
| ECFP4 | 0.729 | 0.792 | 0.867 | 0.822 | 0.799 |
| GraphConv | 0.690 | 0.763 | 0.783 | 0.829 | 0.807 |
| Weave | 0.671 | 0.703 | 0.806 | 0.820 | 0.832 |
| ChemBERTa | 0.643 | 0.622 | - | 0.728 | 0.733 |
| D-MPNN | 0.708 | 0.752 | - | 0.688 | 0.906 |
| CDDD | $0.761 \pm 0.00$ | $0.753 \pm 0.00$ | $0.833 \pm 0.00$ | - | - |
| MolBERT | $0.762 \pm 0.00$ | $0.783 \pm 0.00$ | $0.866 \pm 0.00$ | - | - |
| GraphCL | $0.695 \pm 0.005$ | $0.776 \pm 0.009$ | $0.782 \pm 0.012$ | $0.754 \pm 0.009$ | $0.701 \pm 0.019$ |
| GraphLoG | $0.725 \pm 0.008$ | $0.778 \pm 0.008$ | $0.835 \pm 0.012$ | $0.757 \pm 0.005$ | $0.767 \pm 0.033$ |
| Mol2vec | $0.872 \pm 0.021$ | $0.769 \pm 0.021$ | $0.862 \pm 0.027$ | $0.803 \pm 0.041$ | $0.841 \pm 0.062$ |
| MolR-GCN | $0.890 \pm 0.032$ | $\textbf{0.802} \pm 0.024$ | $\textbf{0.882} \pm 0.019$ | $0.818 \pm 0.023$ | $0.916 \pm 0.039$ |
| MolR-GAT | $0.887 \pm 0.026$ | $0.794 \pm 0.022$ | $0.863 \pm 0.026$ | $\textbf{0.839} \pm 0.039$ | $0.908 \pm 0.039$ |
| MolR-SAGE | $0.879 \pm 0.032$ | $0.793 \pm 0.026$ | $0.859 \pm 0.029$ | $0.811 \pm 0.039$ | $0.890 \pm 0.058$ |
| MolR-TAG | $\textbf{0.895} \pm 0.031$ | $0.801 \pm 0.023$ | $0.875 \pm 0.023$ | $0.820 \pm 0.028$ | $0.913 \pm 0.043$ |

Table 3: Result of AUC in molecule property prediction. The result in the first four blocks is taken from (Honda et al., 2019), (Chithrananda et al., 2020), (Fabian et al., 2020), and (Xu et al., 2021), respectively, while the result in the last two blocks is reported by us. The best results are highlighted in bold, and the best results of baselines are highlighted with underline if MolR is the best.

et al., 2019), *MolBERT* (Fabian et al., 2020), *Mol2vec* (Jaeger et al., 2018), *GraphCL* (You et al., 2020), and *GraphLoG* (Xu et al., 2021). Most baseline results are taken from the literature, while Mol2vec is run by us.

**Experimental Setup**. All datasets are split into training, validation, and test set by 8:1:1. We use the model pretrained on USPTO-479k to process all datasets and output molecule embeddings, then feed embeddings and labels to a Logistic Regression model implemented in scikit-learn (Pedregosa et al., 2011), whose hyperparameters are as default except that solver="liblinear". Each experiment is repeated 20 times, and we report the mean and the standard deviation of *AUC* on the test set.

**Result**. The result of molecule property prediction is reported in Table 3. As we can see, MolR performs the best on 4 out of 5 datasets. We attribute the superior performance of MolR in molecule property prediction to that, MolR is pretrained on USPTO-479k, thus is sensitive to reaction centers according to Proposition 2. Note that reaction centers usually consist of chemically active functional groups, which are essential to determine molecule property.

## 3.3 CHEMICAL REACTION CLASSIFICATION

**Dataset, Baselines, and Experimental Setup**. The goal of chemical reaction classification is to predict the reaction class that a chemical reaction belongs to (e.g., Nitro to amino). We evaluate our model on *USPTO-1k-TPL*, which contains 400,604 training reactions and 44,511 test reactions, and the number of reaction classes is 1,000. We use our pretrained models to calculate the sum of reactant embeddings and the sum of product embeddings in a reaction, then concatenate them as the feature of the reaction. We use an MLP with two 2,048-unit hidden layers as the classifier. Each experiment is repeated 3 times, and we report the mean and standard deviation of *Accuracy* on the test set. We compare our method with *RXNFP* (Schwaller et al., 2021), *AP3-256* (Schneider et al., 2015), and *DRFP* (Probst et al., 2021).

| Methods | Accuracy |
|---|---|
| RXNFP | 0.989 |
| AP3-256-5NN | 0.295 |
| AP3-256-MLP | 0.809 |
| DRFP-5NN | 0.917 |
| DRFP-MLP | 0.977 |
| MolR-GCN | $0.931 \pm 0.022$ |
| MolR-GAT | $0.930 \pm 0.017$ |
| MolR-SAGE | $0.936 \pm 0.025$ |
| MolR-TAG | $0.962 \pm 0.028$ |

Table 4: Result of chemical reaction classification. Result of baselines is taken from (Probst et al., 2021).

**Result**. The result of accuracy is reported in Table 4. 5NN and MLP represent a 5-Nearest-Neighbor and an MLP classifier, respectively. Note that even if these baselines are specially designed to learn reaction fingerprint, our best method MolR-TAG still surpasses AP3-256, and is close to DRFP and RXNFP. This shows that our method can achieve on par performance with state-of-the-art reaction fingerprint learning methods.

## 3.4 GRAPH-EDIT-DISTANCE PREDICTION

Graph-edit-distance (GED) (Gao et al., 2010) is a measure of similarity between two graphs, which is defined as the minimum number of graph edit operations to transform one graph to another. Here we aim to predict the GED between two molecular graphs based on their embeddings. The purpose

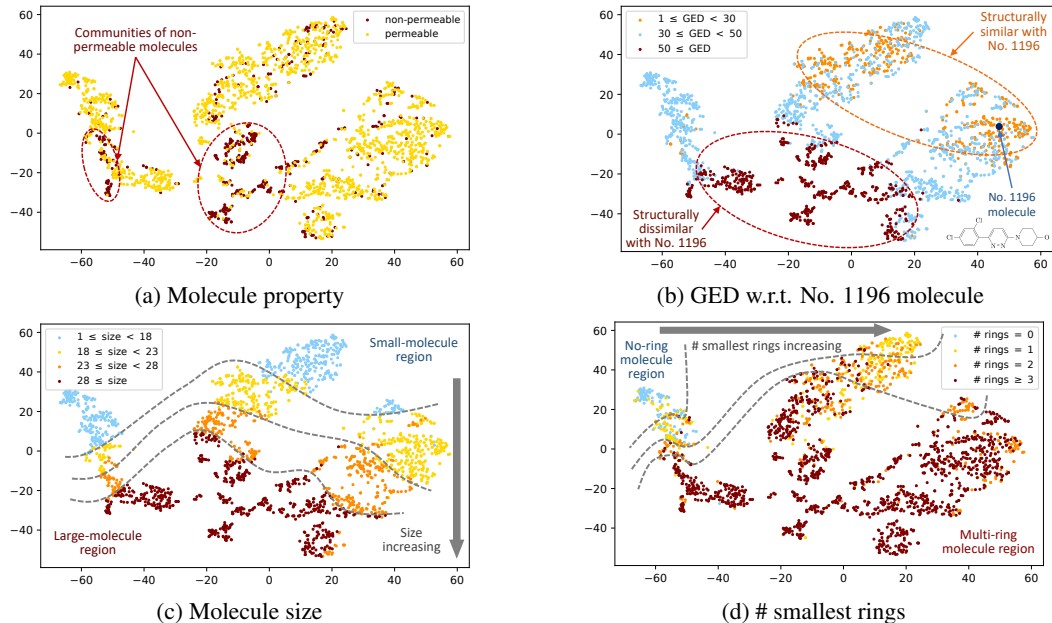

Figure 4: Visualized molecule embedding space for BBBP dataset.

of this task is to show whether the learned molecule embeddings are able to preserve the structural similarity between molecules. Moreover, since calculating the exact GED is NP-hard, the solution to this task can also be seen as an approximation algorithm for GED calculation.

**Dataset, Baselines, and Experimental Setup**. We randomly sample 10,000 molecule pairs from the first 1,000 molecules in *QM9* dataset (Wu et al., 2018), then calculate their ground-truth GEDs using NetworkX (Hagberg et al., 2008). The dataset is split into training, validation, and test set by 8:1:1, and we use our pretrained models to output embeddings for each molecule. The embeddings of a pair of molecules are concatenated or subtracted as the features, and we use a Support Vector Regression model in scikit-learn with default hyperparameters to do the regression. Each experiment is repeated 20 times, and we report the mean and the standard deviation of *RMSE* on the test set. We use *Mol2vec* (Jaeger et al., 2018) and *MolBERT* (Fabian et al., 2020) as baselines.

**Result**. The result of RMSE is reported in Table 2. Our best MolR model reduces the RMSE by 18.5% and 12.8% in concat and subtract mode, respectively, compared with the best baseline. Note that the interval of ground-truth GEDs is $[1, 14]$, whose range is far larger than the RMSE of MolR. The result indicates that MolR can serve as a strong approximation algorithm to calculate GED.

### 3.5 EMBEDDING VISUALIZATION

To intuitively demonstrate the molecule embedding space, we use the pretrained MolR-GCN model to output embeddings of molecules in BBBP dataset, then visualize them using t-SNE (Van der Maaten & Hinton, 2008) shown in Figure 4. In Figure 4a, molecules are colored according to the property of permeability. We find two communities of non-permeable molecules, which demonstrates that MolR can capture molecule property of interest. In Figure 4b, molecules are colored according to their GED to a randomly selected molecule (No. 1196) from BBBP dataset. It is obvious that, molecules that are structurally similar with No. 1196 molecule (colored in orange) are also close to it in the embedding space, while molecules that are structurally dissimilar with No. 1196 molecule (colored in red) are also far from it in the embedding space. The result indicates that MolR can well capture the structural similarity between molecules. In Figure 4c, molecules are colored according to their size (i.e., the number of non-hydrogen atoms). It is clear that the embedding space is perfectly segmented into small-molecule region (upper part) and large-molecule region (lower part). In other words, the vertical axis of the 2D embedding space characterizes molecule size. Last and surprisingly, we find that the horizontal axis are actually related to the number of smallest rings (i.e., rings that do not contain another ring) in a molecule: As shown in Figure 4d, no-ring molecules (colored in blue) are only in the left cluster, one-ring molecules (colored in yellow) are only in the left

and the middle clusters, two-ring molecules (colored in orange) are basically in the middle cluster, while the right cluster mainly consists of molecules with more than 2 rings (colored in red).

We also show that MolR encodes chemical reactions by taking alcohol oxidation and aldehyde oxidation as examples, whose chemical reaction templates are R-CH$_2$OH + O$_2$ $\rightarrow$ R-CHO + H$_2$O and R-CHO + O$_2$ $\rightarrow$ R-COOH, respectively. We first use the pretrained MolR-GCN model to output embeddings for ethanol (CH$_3$CH$_2$OH), 1-octanol (CH$_3$(CH$_2$)$_7$OH), ethylene glycol ((CH$_2$OH)$_2$), as well as their corresponding aldehyde and carboxylic acid, then visualize them using Principle Component Analysis (PCA). The result is visualized in Figure 3, which clearly demonstrates that $h_{\text{CH}_3\text{CHO}} - h_{\text{CH}_3\text{CH}_2\text{OH}} \approx h_{\text{CH}_3(\text{CH}_2)_6\text{CHO}} - h_{\text{CH}_3(\text{CH}_2)_7\text{OH}}$ and $h_{\text{CH}_3\text{COOH}} - h_{\text{CH}_3\text{CHO}} \approx h_{\text{CH}_3(\text{CH}_2)_6\text{COOH}} - h_{\text{CH}_3(\text{CH}_2)_6\text{CHO}}$ (red and orange arrows). Note that the length of blue arrow is about twice as much as the corresponding red or orange arrow, which is exactly because (CH$_2$OH)$_2$/(CH$_2$CHO)$_2$ has two hydroxyl/aldehyde groups to be oxidized.

## 4 Related Work

Existing MRL methods can be classified into two categories. The first is SMILES-based (Fabian et al., 2020; Chithrananda et al., 2020; Honda et al., 2019; Wang et al., 2019; Shin et al., 2019; Zheng et al., 2019b), which use language models to process SMILES strings. For example, MolBERT (Fabian et al., 2020) uses BERT as the base model and designs three self-supervised tasks to learn molecule representation. However, SMILES is 1D linearization of molecular structure and highly depends on the traverse order of molecule graphs. This means that two atoms that are close in SMILES may actually be far from each other and thus not correlated (e.g., the two oxygen atoms in "CC(CCCCCCO)O"), which will mislead language models that rely heavily on the relative position of tokens to provide self-supervision signal.

The second is structure-based methods, which can further be classified into traditional fingerprint-based methods (Rogers & Hahn, 2010; Jaeger et al., 2018) and recent GNN-based methods (Jin et al., 2017; Gilmer et al., 2017; Ishida et al., 2021). For example, Mol2vec (Jaeger et al., 2018) treats molecule substructures (fingerprints) as words and molecules as sentences, then uses a Word2vec-like method to calculate molecule embeddings. However, they cannot identify the importance of different substructures, nor being trained in an end-to-end fashion. GNN-based methods overcome their drawbacks, butt they usually require a large volume of training data due to their complicated architectures, which may limit their generalization ability when the data is sparse.

The idea of taking chemical reactions as prior knowledge is also applied to molecule generation task (Bradshaw et al., 2019; Nguyen & Tsuda, 2021). Compared with these methods which focus on a particular task of molecule generation, the embeddings learned by our model are general-purpose and can benefit a wide range of downstream tasks.

It is worth noticing that our model is also conceptually connected to methods in NLP where embeddings are composable (Bordes et al., 2013; Mikolov et al., 2013). For example, TransE (Bordes et al., 2013) assumes that $\mathbf{h} + \mathbf{r} \approx \mathbf{t}$ for a triplet $(head, relation, tail)$ in knowledge graphs, and Word2vec (Mikolov et al., 2013) learns word embeddings where simple vector addition can produce meaningful results, for example, vec("Germany") + vec("capital") $\approx$ vec("Berlin").

## 5 Conclusion and Future Work

In this work, we use GNNs as the molecule encoder, and use chemical reactions to assist learning molecule representation by forcing the sum of reactant embeddings to be equal to the sum of product embeddings. We prove that our model is able to learn reaction templates that are essential to improve the generalization ability. Our model is shown to benefit a wide range of downstream tasks, and the visualized results demonstrate that the learned embeddings are well-organized and reaction-aware.

We point out four directions as future work. First, environmental condition is also a part of chemical reactions that we will consider to model. Second, as discussed in Section 2.2, it is worth studying how to explicitly output learned reaction templates. Third, it is interesting to investigate how to differentiate stereoisomers in the embedding space, since our model (as well as baselines) cannot handle stereoisomerism. Last, including additional information (e.g., textual description of molecules) to assist learning molecule representation is also a promising direction.

ACKNOWLEDGEMENT

This research is based upon work supported by the Molecule Maker Lab Institute: an AI research institute program supported by NSF under award No. 2019897 and No. 2034562. The views and conclusions contained herein are those of the authors and should not be interpreted as necessarily representing the official policies, either expressed or implied, of the U.S. Government. The U.S. Government is authorized to reproduce and distribute reprints for governmental purposes notwithstanding any copyright annotation therein.

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

## A    Detailed Description of GNN Architectures

Before applying GNNs, we first use pysmiles[3] to parse the SMILES strings of molecules to NetworkX graphs. Then we use the following GNNs as our molecule encoder and investigate their performance in experiments. The implementation of GNNs is based on Deep Graph Library (DGL)[4]. For all GNNs, the number of layers is 2, and the output dimension of all layers is 1,024. The activation function $\sigma$ is identity for the last layer, and ReLU for all but the last layer. We keep a bias term $b$ for all the linear transformations below as it is the default setting in DGL, but our experiments show that the bias term can barely affect the result, so it is not shown in the following equations for clarity.

**Graph Convolutional Networks (GCN)** (Kipf & Welling, 2017). In GCN, AGGREGATE is implemented as weighted average with respect to the inverse of square root of node degrees:

$$h_i^k = \sigma\Big(W^k \sum\nolimits_{j \in \mathcal{N}(i) \cup \{i\}} \alpha_{ij} h_j^{k-1}\Big), \tag{6}$$

where $W^k$ is a learnable matrix, $\alpha_{ij} = 1/\sqrt{|\mathcal{N}(i)| \cdot |\mathcal{N}(j)|}$, and $\sigma$ is activation function.

**Graph Attention Networks (GAT)** (Veličković et al., 2018). In GAT, AGGREGATE is implemented as multi-head self-attention:

$$h_i^k = \mathop{\big\|}_{s=1}^{S} \sigma\Big(W^{k,s} \sum\nolimits_{j \in \mathcal{N}(i) \cup \{i\}} \mathrm{SOFTMAX}(\alpha_{ij}^{k,s}) h_j^{k-1}\Big), \tag{7}$$

where $\alpha_{ij}^{k,s} = \mathrm{LeakyReLU}\Big({w^{k,s}}^\top \big[W^{k,s} h_i^k \,\|\, W^{k,s} h_j^k\big]\Big)$ is (unnormalized) attention weight, $w$ is a learnable vector, $\|$ denotes concatenate operation, and $S$ is the number of attention heads. In our experiments, $S$ is set as 16 and the dimension of each attention head is set as 64, so that the total output dimension is still 1,024.

**Graph Sample and Aggregate (GraphSAGE)** (Hamilton et al., 2017). In the pooling variant of GraphSAGE, AGGREGATE is implemented as:

$$h_i^k = \sigma\Big(W^{k,1}\big(h_i^{k-1} \,\big\|\, h_{\mathcal{N}(i)}^k\big)\Big), \tag{8}$$

where $h_{\mathcal{N}(i)}^k = \mathrm{MAX}\Big(\big\{\mathrm{ReLU}(W^{k,2} h_j^{k-1})\big\}_{j \in \mathcal{N}(i)}\Big)$ is aggregated neighborhood representation, and $\mathrm{MAX}$ is an element-wise max-pooling function. In addition to MAX, AGGREGATE can also be implemented as MEAN, GCN, and LSTM, but our experiments show that MAX achieves the best performance.

**Topology Adaptive Graph Convolutional Networks (TAGCN)** (Du et al., 2017). In TAGCN, if we use $H^k$ to denote the representation matrix of all atoms in layer $k$, then AGGREGATE can be written as:

$$H^k = \sigma\Big(\sum\nolimits_{l=0}^{L} \tilde{A}^l H^{k-1} W^{k,l}\Big), \tag{9}$$

where $\tilde{A} = D^{-1/2} A D^{-1/2}$ is the normalized adjacency matrix, and $L$ is the size of the local filter. We set $L$ to 2 in experiments. It should be noted that, different from other GNNs, a single TAGCN layer can aggregate node information from neighbors up to $L$ hops away. Therefore, the actual number of layers for TAGCN is $L^K$, which is $2^2 = 4$ in our experiments.

## B    Proof of Proposition 1

*Proof.*    To prove that "$\rightarrow$" is an equivalence relation on $2^M$, we need to prove that "$\rightarrow$" satisfies reflexivity, symmetry, and transitivity:

*Reflexivity*. For any $A \in 2^M$, it is clear that $\sum_{i \in A} h_i = \sum_{i \in A} h_i$. Therefore, we have $A \rightarrow A$.

---

[3] https://pypi.org/project/pysmiles/
[4] https://www.dgl.ai/

*Symmetry*. For any $A, B \in 2^M$, $A \to B \Rightarrow \sum_{i \in A} h_i = \sum_{i \in B} h_i \Rightarrow B \to A$, and vice versa. Therefore, we have $A \to B \Leftrightarrow B \to A$.

*Transitivity*. If $A \to B$ and $B \to C$, then we have $\sum_{i \in A} h_i = \sum_{i \in B} h_i = \sum_{i \in C} h_i$. Therefore, we have $A \to C$. $\qquad\square$

## C  PROOF OF PROPOSITION 2

*Proof.*  Since the READOUT function is summation, the embedding of molecule $G = (V, E)$ is the sum of final embeddings of all atoms that $G$ consists of: $h_G = \sum_{a_i \in V} h_i^K$. Therefore, for the reaction $R \to P$, we have $\sum_{r \in R} h_r - \sum_{p \in P} h_p = \sum_{r \in R} \sum_{v \in r} h_v^K - \sum_{p \in P} \sum_{v \in p} \tilde{h}_v^K$, where $h_v^K$ is the representation vector of atom $v$ calculated by a $K$-layer GNN on the reactant graph, and $\tilde{h}_v^K$ is the representation vector of atom $v$ calculated by the same $K$-layer GNN on the product graph. If we treat all reactants $R$ (all products $P$) in a chemical reaction as one graph where each reactant (product) is a connected component of this graph, the above equation can be written as $\sum_{v \in R} h_v^K - \sum_{v \in P} \tilde{h}_v^K$. Denote $C^k$ as the set of atoms whose distance from the reaction center $C$ is $k$ ($C^0 = C$). We can prove, by induction, that $\sum_{v \in R} h_v^K - \sum_{v \in P} \tilde{h}_v^K = \sum_{v \in \cup_{k=0}^{K-1} C^k} h_v^K - \sum_{v \in \cup_{k=0}^{K-1} C^k} \tilde{h}_v^K$:

Let's first consider the initial case where $K = 1$. Then the final representation of an atom depends on the initial feature of itself as well as its one-hop neighbors. Therefore, atoms whose final representation differs from $R$ to $P$ must be those who have at least one bond changed from $R$ to $P$, which are exactly the atoms in the reaction center $C$. Representation of atoms that are not within the reaction center remains the same before and after reaction, and thus are eliminated. Therefore, we have $\sum_{v \in R} h_v^1 - \sum_{v \in P} \tilde{h}_v^1 = \sum_{v \in C} h_v^1 - \sum_{v \in C} \tilde{h}_v^1$.

Induction step. Suppose we have $\sum_{v \in R} h_v^K - \sum_{v \in P} \tilde{h}_v^K = \sum_{v \in \cup_{k=0}^{K-1} C^k} h_v^K - \sum_{v \in \cup_{k=0}^{K-1} C^k} \tilde{h}_v^K$ for $K \geq 1$. This means that representation of atoms whose distance from the reaction center is no larger than $K - 1$ becomes different from $R$ to $P$, while representation of atoms whose distance from the reaction center is larger than $K - 1$ remains the same. Then for $K + 1$, the number of GNN layers increases by one. This means that atoms adjacent to the set $\cup_{k=0}^{K-1} C^k$ are additionally influenced, and their representations are also changed. It is clear that the set of atoms whose representation is changed is now $\cup_{k=0}^{K} C^k$. Therefore, we have $\sum_{v \in R} h_v^{K+1} - \sum_{v \in P} \tilde{h}_v^{K+1} = \sum_{v \in \cup_{k=0}^{K} C^k} h_v^{K+1} - \sum_{v \in \cup_{k=0}^{K} C^k} \tilde{h}_v^{K+1}$ for $K + 1$.

Since $\sum_{r \in R} h_r - \sum_{p \in P} h_p = \sum_{v \in \cup_{k=0}^{K-1} C^k} h_v^K - \sum_{v \in \cup_{k=0}^{K-1} C^k} \tilde{h}_v^K$, we can conclude that the residual term $\sum_{r \in R} h_r - \sum_{p \in P} h_p$ is a function of $h_a^K$ if and only if $a \in \cup_{k=0}^{K-1}$, i.e., the distance between atom $a$ and reaction center $C$ is less than $K$. $\qquad\square$

## D  HYPERPARAMETER SENSITIVITY

We investigate the sensitivity of MolR-GCN in the task of reaction product prediction to the following hyperparameters: the number of GCN layers $K$, the dimension of embedding, the margin $\gamma$, the batch size, and the implementation of READOUT function. Since the impact of the first three hyperparameters are highly correlated, we report the result of MolR-GCN when jointly influenced by two hyperparameters.

Figure 5a shows the result of MolR-GCN when varying the number of GCN layers and the dimension of embedding. It is clear that MolR-GCN does not perform well when $K$ is too small ($K = 1$) or too large ($K = 3, 4$), which empirically justifies our claim in Remark 3 in Section 2.2 that the number of GNN layers $K$ can greatly impact the learned reaction templates as well as the performance of our model.

Figure 5b demonstrates that the margin $\gamma$ can greatly influence the model performance as well, and such influence is highly related to the dimension of embedding: When the embedding dimension increases from 128 to 2,048, the optimal $\gamma$ also increases from 2 from 8. This is because the expected Euclidean distance between two vectors will increase proportionally to the square root of the vector length, which forces the optimal $\gamma$ to move right accordingly.

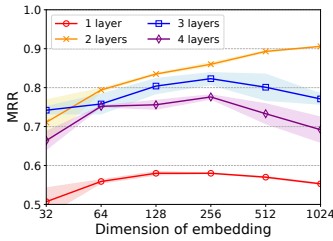
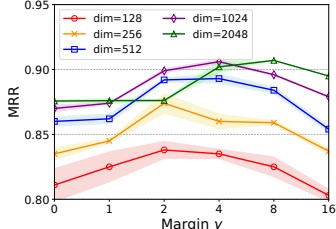
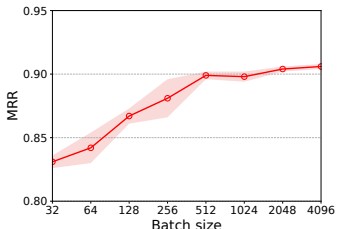

(a) Sensitivity to # GCN layers and the dimension of embedding

(b) Sensitivity to the dimension of embedding and the margin $\gamma$

(c) Sensitivity to the batch size

Figure 5: Sensitivity of MolR-GCN to the number of GCN layers, the dimension of embedding, the margin $\gamma$, and the batch size.

| Batch size | 64 | 128 | 256 | 512 | 1,024 | 2,048 | 4,096 |
|---|---|---|---|---|---|---|---|
| Time cost per epoch (s) | $198.0 \pm 13.8$ | $145.7 \pm 10.4$ | $141.3 \pm 0.9$ | $132.2 \pm 9.1$ | $118.1 \pm 5.9$ | $113.3 \pm 3.0$ | $110.4 \pm 1.5$ |
| Max memory cost (GiB) | 5.3 | 5.7 | 5.8 | 6.2 | 7.5 | 10.5 | 14.8 |

Table 5: The average time cost per epoch and the maximal memory cost of MolR-GCN when varying the batch size.

Figure 5c shows that when batch size increases from 32 to 4,096, MRR also increases from 0.831 to 0.906. This is because a larger batch size will introduce more unmatched reactant-product pairs as negative samples, which provides more supervision for model optimization. In the extreme case of batch_size = # training instances, all pairwise molecule distances will be calculated and optimized. Moreover, as shown in Table 5, the time cost per epoch can also be significantly reduced when increasing the batch size (run on an NVIDIA V100 GPU). However, a larger batch size will also require more memory usage of a minibatch in GPU. As shown in Table 5, when batch size increases fro 64 to 4,096, the required memory of MolR-GCN also increases from 5.3 GiB to 14.8 GiB. This prevents us to further increase batch size since the memory of NVIDIA V100 GPU is 16 GiB.

Table 6 reports the result of MolR-GCN with different implementations of the READOUT function: *AvgPooling*, which applies average pooling over the atoms in a molecule graph; *MaxPooling*, which applies max pooling over the atoms in a molecule graph; *Attention*, which uses an MLP to take the embedding of an atom as input and outputs its importance score, then averages all atom embeddings weighted by their importance scores to obtain the molecule embedding; *SortPooling* (Zhang et al., 2018), which sorts atom embeddings before calculating a molecule embedding; *Set2set* (Vinyals et al., 2016), which is also an attention-like method that iteratively calculates a query vector and uses the query vector to calculate the weighted average of atom embeddings; *SetTransformer* (Lee et al., 2019), which is another attention-based method that calculates the attention of atoms using Transformer. We also attach the result of *SumPooling* here, which is exactly what we use in our model. The result shows that SumPooling significantly outperforms all other pooling functions, even the ones that are really complicated. This is because, as shown in Proposition 2, our model can learn reaction templates with SumPooling, which significantly improves the generalization ability.

## E ABLATION STUDY

To demonstrate that introducing the constraint of chemical reaction equivalence can indeed benefit downstream tasks, we conduct ablation study on MolR in the task of molecule property predic-

| Metrics | MRR | MR | Hit@1 | Hit@3 | Hit@5 | Hit@10 |
|---|---|---|---|---|---|---|
| AvgPooling | $0.777 \pm 0.001$ | $172.2 \pm 2.9$ | $0.730 \pm 0.001$ | $0.809 \pm 0.002$ | $0.832 \pm 0.002$ | $0.858 \pm 0.001$ |
| MaxPooling | $0.730 \pm 0.001$ | $451.3 \pm 10.0$ | $0.675 \pm 0.001$ | $0.771 \pm 0.001$ | $0.793 \pm 0.001$ | $0.816 \pm 0.001$ |
| Attention | $0.669 \pm 0.009$ | $242.9 \pm 7.8$ | $0.597 \pm 0.012$ | $0.719 \pm 0.007$ | $0.753 \pm 0.005$ | $0.788 \pm 0.005$ |
| Sortpooling | $0.380 \pm 0.032$ | $1210.7 \pm 147.8$ | $0.279 \pm 0.031$ | $0.433 \pm 0.037$ | $0.498 \pm 0.035$ | $0.575 \pm 0.028$ |
| Set2Set | $0.731 \pm 0.000$ | $457.9 \pm 0.0$ | $0.649 \pm 0.000$ | $0.790 \pm 0.000$ | $0.831 \pm 0.000$ | $0.873 \pm 0.000$ |
| SetTransformer | $0.700 \pm 0.033$ | $301.9 \pm 59.7$ | $0.632 \pm 0.036$ | $0.743 \pm 0.026$ | $0.772 \pm 0.023$ | $0.804 \pm 0.019$ |
| SumPooling | $0.905 \pm 0.001$ | $34.5 \pm 2.4$ | $0.867 \pm 0.001$ | $0.938 \pm 0.001$ | $0.950 \pm 0.001$ | $0.961 \pm 0.002$ |

Table 6: Sensitivity of MolR-GCN to the READOUT function.

| Datasets | BBBP | HIV | BACE | Tox21 | ClinTox | USPTO-1k-TPL | QM9 |
|---|---|---|---|---|---|---|---|
| GCN | $0.883 \pm 0.036$ | $0.781 \pm 0.024$ | $0.858 \pm 0.026$ | $0.825 \pm 0.037$ | $0.865 \pm 0.041$ | $0.728 \pm 0.029$ | $0.985 \pm 0.021$ |
| MolR-GCN | $0.890 \pm 0.032$ | $0.802 \pm 0.024$ | $0.882 \pm 0.019$ | $0.818 \pm 0.023$ | $0.916 \pm 0.039$ | $0.931 \pm 0.022$ | $0.976 \pm 0.026$ |
| GAT | $0.893 \pm 0.041$ | $0.790 \pm 0.008$ | $0.843 \pm 0.033$ | $0.805 \pm 0.024$ | $0.897 \pm 0.019$ | $0.728 \pm 0.033$ | $1.015 \pm 0.024$ |
| MolR-GAT | $0.887 \pm 0.026$ | $0.794 \pm 0.022$ | $0.863 \pm 0.026$ | $0.839 \pm 0.039$ | $0.908 \pm 0.039$ | $0.930 \pm 0.017$ | $1.007 \pm 0.021$ |
| SAGE | $0.855 \pm 0.021$ | $0.814 \pm 0.006$ | $0.832 \pm 0.022$ | $0.827 \pm 0.048$ | $0.884 \pm 0.056$ | $0.729 \pm 0.037$ | $1.102 \pm 0.023$ |
| MolR-SAGE | $0.879 \pm 0.032$ | $0.793 \pm 0.026$ | $0.859 \pm 0.029$ | $0.811 \pm 0.039$ | $0.890 \pm 0.058$ | $0.936 \pm 0.025$ | $0.918 \pm 0.028$ |
| TAG | $0.859 \pm 0.013$ | $0.791 \pm 0.023$ | $0.859 \pm 0.028$ | $0.815 \pm 0.040$ | $0.900 \pm 0.022$ | $0.677 \pm 0.023$ | $0.969 \pm 0.019$ |
| MolR-TAG | $0.895 \pm 0.031$ | $0.801 \pm 0.023$ | $0.875 \pm 0.023$ | $0.820 \pm 0.028$ | $0.913 \pm 0.043$ | $0.962 \pm 0.028$ | $0.960 \pm 0.027$ |

Table 7: Ablation study of MolR in the task of molecule property prediction (AUC), chemical reaction classification (Accuracy), and graph-edit-distance prediction (RMSE), w/o imposing the constraint of chemical reaction equivalence.

tion, chemical reaction classification, and graph-edit-distance prediction. We directly train the four GNNs, i.e., GCN, GAT, SAGE, and TAG, on these datasets, then report their results in Table 7. For molecule property prediction task, our method can improve the AUC by $1\% \sim 5\%$ compared with the GNN in most cases. For chemical reaction classification task, we observe a huge improvement of our models against GNNs ($20\% \sim 28\%$), of which the reason may be that our model is pretrained on USPTO dataset as well. For GED prediction task, the improvement of our method is rather significant when using SAGE as the encoder (around $18\%$), and the improvement is around $1\%$ for the rest three GNNs. The result of the ablation study shows that the constraint of chemical reaction equivalence can effectively improve the quality of learned molecule embeddings in various downstream tasks.

# F    DETAILS OF MULTI-CHOICE QUESTIONS ON PRODUCT PREDICTION

Our multi-choice questions on product prediction are collected from online resources of Oxford University Press,[5,6] mhpraticeplus.com[7], and GRE Chemistry Test Practice Book[8]. We filter out the questions 1) whose choices contain stereoisomers, since our model as well as baselines cannot handle stereoisomerism, and 2) that contain molecules that cannot be successfully parsed by our method or baselines. We manually convert the figures of molecules into SMILES strings. The first five questions are listed below as examples (the correct answer is given after the question), while the complete SMILES version of these questions are included in the code repository.

Examples $1 \sim 5$:

1. 

$$\xrightarrow[\text{2. } H_2O, H^+]{\text{1. } CH_3MgBr}$$

Which of the following is the major product of the reaction shown above? (D)

---

[5] https://global.oup.com/uk/orc/chemistry/okuyama/student/mcqs/ch19/
[6] https://global.oup.com/uk/orc/chemistry/okuyama/student/mcqs/ch21/
[7] https://www.mhpracticeplus.com/mcat_review/MCAT_Review_Smith.pdf
[8] https://www.ets.org/s/gre/pdf/practice_book_chemistry.pdf

(E) 
$$\text{CHO}$$
$$\text{CH}_3$$

2. $CH_3CH_2CH_2COOH \xrightarrow[\text{2. } H_3O^+]{\text{1. LiAlH}_4 \text{ (excess)}}$

Which of the following is the major organic product of the reaction shown above? (D)
(A) $CH_3CH_2CH_2CH(OH)_2$
(B) $CH_3CH_2CH_2CHO$
(C) $CH_3CH_2CH_2CH_2OH$
(D) $CH_3CH_2CH_2CH_3$
(E) $CH_3CH_2C{\equiv}CH$

3. 
$$\text{OCH}_3$$
$$\xrightarrow{\text{HI}}$$

Which of the following are the major products of the reaction shown above? (D)

(A) ⬡ + $CH_3OH$

(B) ⬡OH + $CH_4$

(C) ⬡I + $CH_3OH$

(D) ⬡OH + $CH_3I$

(E) ⬡ + $CH_3I$

4. 
$$\text{O} \quad \text{O}$$
$$\text{O} \quad \text{CH}_3$$
$$\xrightarrow{H_3O^+}$$

Which of the following are the products of the reaction shown above? (E)

(A) ⬡CHO + $CH_3CHO$

(B) ⬡COOH + $CH_3CH_2OH$

(C) ⬡CH$_2$OH + $CH_3CH_2OH$

(D) ⬡CHO + $CH_3COOH$

(E) ⬡COOH + $CH_3COOH$

5.

What is the product of the reaction shown above for *para*-cresol? (A)

(A)

(B)

(C)

(D)

(E)

## G    CASE STUDY ON USPTO-479K DATASET

The case study on USPTO-479k dataset is shown in Table 8. To avoid cherry-picking, we select the first 20 reaction instances (No. 0 ∼ No. 19) in the test set for case study. In Table 8, the first column is the index of reactions, the second and the third column are the reactant(s) and the ground-truth product of this reaction, respectively, and the fourth to sixth column are the product predicted by MolR-GCN, Mol2vec, and MolBERT, respectively. Reactions whose product is correctly predicted by all the three methods are omitted for clarity. The index below a predicted product indicates the actual reaction that this product comes from.

As shown in Table 8, MolR-GCN gives a wrong answer on only 2 reactions, while Mol2vec and MolBERT make mistakes on 6 reactions. Moreover, for the two reactions that MolR-GCN does not answer correctly (No. 5 and No. 10), Mol2vec and MolBERT give the exactly same wrong answers as well, which shows that the product of the two reactions are indeed very hard to predict. Actually, for the No. 5 reaction, the predicted product (No. 32353) is identical to the reactant, which, technically speaking, cannot be seen as a wrong answer. For the No. 10 reaction, the predicted product (No. 18889) is very similar to the ground-truth product and they only differ in one atom (N vs S).

Another two examples that are worth mentioning are: (1) The No. 6 reaction, where the first reactant contains a triangle ring but it breaks up in the product. MolR-GCN correctly predicts the product, but Mol2vec and MolBERT make mistakes by simply combining the two reactants together. (2) The No. 17 reaction, where the nitro group ($-NO_2$) disappears after the reaction. MolR-GCN predicts such chemical change successfully, but Mol2vec and MolBERT still retain the nitro group in their predicted products.

| No. | Reactant(s) | Ground-truth product | Predicted product by MolR-GCN | Predicted product by Mol2vec | Predicted product by MolBERT |
|---|---|---|---|---|---|
| 5 | | | (No. 32353) | (No. 32353) | (No. 32353) |
| 6 | | | Same as ground-truth | (No. 39181) | (No. 24126) |
| 8 | | | Same as ground-truth | (No. 11233) | (No. 17526) |
| 10 | | | (No. 18889) | (No. 18889) | (No. 18889) |
| 13 | | | Same as ground-truth | Same as ground-truth | (No. 37024) |
| 16 | | | Same as ground-truth | (No. 21947) | Same as ground-truth |
| 17 | | | Same as ground-truth | (No. 2029) | (No. 22247) |

Table 8: Case study on the first 20 reaction instances (No. 0 ~ No. 19) in the test set of USPTO-479k. Reactions whose product is correctly predicted by all the three methods are omitted for clarity. The index below a predicted product indicates the actual reaction that this product comes from.

