# OpenReview forum: "Chemical-Reaction-Aware Molecule Representation Learning"
_ICLR.cc/2022/Conference — ICLR 2022 Poster_

### Official Review · Reviewer_3hy7 · 2021-11-01

**Correctness:** 3
**Technical Novelty And Significance:** 4
**Empirical Novelty And Significance:** 3
**Recommendation:** 6
**Confidence:** 4

**Main Review:**

Strength
1.	The idea of using chemical reactions to perform contrastive learning for molecular representation is novel, which offers alternative solutions to the problem by leveraging domain knowledge from the synthetic perspective. This formulation avoids the unjustified masking of molecular graphs in previous works (as a small modification of molecular structure may result in a series of changes in chemical properties). The provided benchmarks open a new avenue for the molecular contrastive learning community.
2.  The paper is generally well written with minor typos.

Weakness
1.	Some of the experiments' decisions make the experimental results are not convincing for me: 1) taking R->P as positive samples potentially cause ambiguity. For example, if there are two reactions, OHCH2CHOHCH2OH -> CH2CHOHCH2OH + OH and OHCH2CHOHCH2OH -> OHCH2CHCH2OH, the MolR would force h_{OHCH2CHCH2OH} and h_{CH2CHOHCH2OH} to be similar. This is unreasonable as the position of a functional group will influence their molecular properties significantly.
2.	The authors did not clarify what kind of splits are used for training/validation/test split. If the molecular properties data are randomly split, the results of D-MPNN are significantly lower than the values reported in its original paper and other SOTA GNN models (e.g.[1][2][3]).
3.	The authors should report the accuracy of the original methods (GCN, GAT, SAGE, TAG); therefore, we can make a clear comparison of how many improvements we can expect by using MolR
4.	Other graph contrastive learning methods (GraphCL[4], GraphLoG[5]) should also be compared.

Reference
[1] Xiong Z, Wang D, Liu X, et al. Pushing the boundaries of molecular representation for drug discovery with the graph attention mechanism[J]. Journal of medicinal chemistry, 2019, 63(16): 8749-8760.

[2] Li Z, Yang S, Song G, et al. Conformation-Guided Molecular Representation with Hamiltonian Neural Networks[C]//International Conference on Learning Representations. 2020.

[3] Song Y, Zheng S, Niu Z, et al. Communicative Representation Learning on Attributed Molecular Graphs[C]//IJCAI. 2020: 2831-2838.

[4] You Y, Chen T, Sui Y, et al. Graph contrastive learning with augmentations[J]. Advances in Neural Information Processing Systems, 2020, 33: 5812-5823.

[5] Xu M, Wang H, Ni B, et al. Self-supervised Graph-level Representation Learning with Local and Global Structure[J]. arXiv preprint arXiv:2106.04113, 2021.


**Summary Of The Paper:**

This paper proposes to use chemical reactions to pretrain molecular graph representations by 1) considering the reactants and products to be positive samples in the contrastive learning 2) pretraining using USPTO-479k reaction dataset.  The result is a pertaining method that outperforms a variety of baseline methods on reaction prediction and molecular property prediction tasks.

**Summary Of The Review:**

The motivation for this research area is strong and there have been several papers addressing similar problems at previous top-tier machine learning conferences. I think ICLR is a good venue for this manuscript.

---

> ### Author Response · Authors · 2021-11-18
> **Response to Reviewer 3hy7**
>
> Thank you so much for your valuable comments. We have carefully revised our paper according to your advice. Our response to your questions are as follows:
>
> ================================================================
>
> Q1: If there are two reactions, OHCH2CHOHCH2OH -> CH2CHOHCH2OH + OH and OHCH2CHOHCH2OH -> OHCH2CHCH2OH + OH, the MolR would force h_{OHCH2CHCH2OH} and h_{CH2CHOHCH2OH} to be similar. This is unreasonable as the position of a functional group will influence their molecular properties significantly.
>
> > Given the example you mentioned, it is indeed the case that our model will make h_{OCCCO} and h_{CC(O)CO} similar. Actually, we think that their embeddings should be similar, because OCCCO and CC(O)CO are very similar in terms of structure.
> >
> >It is true that the position of a functional group will influence the molecular property, and the properties of OCCCO and CC(O)CO are not exactly the same. Specifically, OCCCO is a continuous three carbon chain with hydroxy groups on both ends, so it has two primary alcohol, and CC(O)CO also has a three carbon chain but with an extra hydroxy group on the second carbon, it still contains a primary alcohol with another secondary alcohol. This minor structural difference results in slightly different reaction mechanism, one goes through the E2 mechanism with its primary alcohol, the other one results in parallel reactions that include both E1 and E2 reactions because of its primary and secondary alcohols. This difference is small enough that most of the products are similar due to the dominant E2 reactions.
> >
> >This example shows that they should behave similarly during reactions with the exact same functional groups . Thus, these two should have relatively similar embedding, while compared to the much larger difference, for example, the difference between h_{OCCCO} and h_{C1=CC=CC=C1} (benzene). From this perspective, it is totally fine that the embeddings of OCCCO and CC(O)CO are close to each other.
>
> ================================================================
>
> Q2: The authors did not clarify what kind of splits are used for training/validation/test split. If the molecular properties data are randomly split, the results of D-MPNN are significantly lower than the values reported in its original paper and other SOTA GNN models.
>
> > The molecule properties data are randomly split. The result of D-MPNN is directly taken from Table 1 in Chithrananda et al. [1]. We looked at the values reported in the original paper of D-MPNN. They have so many versions of results, and the results are all shown in figures rather than tables, which makes it really hard to get the exact number. We think that Figure 19(b) in their paper can represent the best results of D-MPNN. According to Figure 19(b), the AUC scores of D-MPNN on BBBP, HIV, BACE, Tox21, and ClinTox are around 0.9, 0.8, 0.85, 0.8, and 0.9, respectively, which are indeed lower than the values reported in [1], but are still not as good as our model in most cases.
> >
> > [1] Chithrananda et al. “Chemberta: Large-scale self-supervised pretraining for molecular property prediction.” arXiv preprint arXiv:2010.09885 (2020).
>
> ================================================================
>
> Q3: The authors should report the accuracy of the original methods (GCN, GAT, SAGE, TAG); therefore, we can make a clear comparison of how many improvements we can expect by using MolR.
>
> > We followed your advice and added the ablation study where the constraint of chemical reaction equivalence is removed, and we directly train the GNNs for downstream tasks. Please refer to Appendix E and Table 6 for details.
>
> ================================================================
>
> Q4: Other graph contrastive learning methods (GraphCL, GraphLoG) should also be compared.
>
> > We have added these two methods as our baselines. Please refer to Table 3 on page 7 for details.

---

### Official Review · Reviewer_LLpm · 2021-11-02

**Correctness:** 4
**Technical Novelty And Significance:** 3
**Empirical Novelty And Significance:** 3
**Recommendation:** 6
**Confidence:** 3

**Main Review:**

The strength: the idea of the paper using the reaction to embed molecules in a space where the summation of reactant embeddings is equal to the summation of product embeddings is very interesting. And the experimental results also seem to be promising. I just have a few concerns as follows:

1.  "The reaction center of R → P is defined as an induced subgraph of reactants R, in which each atom has
at least one bond whose type differs from R to P (“no bond” is also seen as a bond type)" this sounds bit confusing, I did not really get what does it mean, I can understand in terms of graph edit distance, "a reaction center is a minimal set of graph edits needed to transform reactants to products." but still very confused.

2. It is not clear to me how the current model can learn the reaction template.

3. figure 3 is not really clear to me


**Summary Of The Paper:**

The paper is about learning a vector representation of molecules in a way that the learned representation preserves the equivalence of molecules with respect to chemical reactions. They do so by forcing the sum of reactant embeddings and the sum of product embeddings to be equal for each chemical equation.  They have also shown experimentally that such embeddings can improve the performance of downstream tasks such as chemical reaction prediction, molecule property prediction, and graph edit distance prediction problems.

**Summary Of The Review:**

The idea presented in the paper is novel, interesting, simple, and direct.  The results seem to be promising.

---

> ### Author Response · Authors · 2021-11-18
> **Response to Reviewer LLpm**
>
> Thank you so much for your valuable comments. We have carefully revised our paper according to your advice. Our response to your questions are as follows:
>
> ================================================================
>
> Q1: “The reaction center of R → P is defined as an induced subgraph of reactants R, in which each atom has at least one bond whose type differs from R to P (“no bond” is also seen as a bond type)” this sounds a bit confusing.
>
> > We are sorry for not making this clear. We have revised the definition of a reaction center. Please refer to the highlighted part on page 4 for details.
>
> ================================================================
>
> Q2: It is not clear to me how the current model can learn the reaction template.
>
> > We give an example here to make this more clear and intuitive. Suppose the training data contains a reaction: CH3CH2COOH + CH3CH2CH2OH -> CH3CH2COOCH2CH2HC3 + H2O, and suppose that we are using a 3-layer GNN with SumPooling as the readout function. According to our training objective, we are trying to minimize the term: h(CH3CH2COOH) + h(CH3CH2CH2OH) - h(CH3CH2COOCH2CH2HC3) - h(H2O). If we use SMILES to represent molecules, the term becomes h(CCC(=O)O) + h(CCCO) - h(CCC(=O)OCCC) - h(O). If we represent each atom with a unique letter, then the term becomes h(abc(=d)e) + h(fghi) - h(jkl(=m)nopq) - h(r). Since we are using sum as the readout function, a molecule embedding is simply the sum of all the final embeddings of its contained atoms, this means that we are minimizing the term: h(a) + … + h(i) - h(j) - … - h(r).
> >
> > Now here comes the key part: since we are using a 3-layer GNN to learn the final embedding of atoms, if the set of the 3-hop neighbors of an atom remains the same before and after the reaction, its final embedding will also remain unchanged. It is obvious to see that the only two atoms whose 3-hop neighbors remain unchanged are atom a and atom f, whose corresponding atom after the reaction is atom j and q, respectively. This means that h(a) = h(j), and h(f) = h(q). Therefore, a-j pair and f-q pair are eliminated from the above term, and we are actually minimizing T = h(b) + h(c) + h(d) + h(e) + h(g) +h(h) + h(i) - h(k) - h(l) - h(m) - h(n) - h(o) - h(p) - h(r).
> >
> > Suppose that we have a new reaction in the test set R1-CH2COOH + R2-CH2CH2OH -> R1-CH2COOCH2CH2-R2 + H2O, where R1 and R2 can be any functional group. The residual term of this reaction is h(R1-CC(=O)O) + h(R2-CCO) - h(R1-CC(=O)OCC-R2) -h(O), which can be written as h(R1-bc(=d)e) + h(R2-ghi) - h(R1-kl(=m)nop-R2) - h(r). It is obvious that the 3-hop neighbor set for atoms in R1 and R2 remain unchanged before and after the reaction, which means that the final embeddings of atoms in R1 and R2 remain the same before and after the reaction, and they are actually eliminated from the residual term. Therefore, the residual term for this new reaction is actually h(b) + h(c) + h(d) + h(e) + h(g) +h(h) + h(i) - h(k) - h(l) - h(m) - h(n) - h(o) - h(p) - h(r), which is the same as the above T.
> >
> > If our model is perfectly trained and the residual term T=0, then for the above new reaction, the residual term will also be 0. This means that our model will know that the new reaction holds for sure.
> >
> > To conclude, our model will eliminate most of the common parts of reactants and products, and only consider the atoms in the reaction center or close to the reaction center. This is how our model can learn reaction templates and achieve high generalization ability.
>
> ================================================================
>
> Q3: Figure 3 is not really clear to me.
>
> > Let’s only look at the red circle (CH3CH2OH), red triangle (CH3CHO), yellow circle (CH3(CH2)7OH), and yellow triangle (CH3(CH2)6CHO) in Figure 3. We know that CH3CH2OH + O2 - > CH3CHO + H2O, and CH3(CH2)7OH + O2 -> CH3(CH2)6CHO + H2O, which means that in our model, it should be the case that h(red circle) + h(O2) = h(red triangle) + h(H2O), and h(yellow circle) + h(O2) = h(yellow triangle) + h(H2O). This means that h(red triangle) - h(red circle) = h(O2) - h(H2O) = h(yellow triangle) - h(yellow circle), which means that the vector from red circle to red triangle should be the same as the vector from yellow circle to yellow triangle. We can verify by Figure 5 that it is indeed true.

---

> ### Comment · Reviewer_LLpm · 2021-11-29
> **comments on author response**
>
> Thank you for the author's clarification on the definition of reaction center, it is very clear now. Thank you for the response to other questions too. After reading the author's response and other reviewers' comments I decided to keep my original score.

---

### Official Review · Reviewer_17vx · 2021-11-02

**Correctness:** 4
**Technical Novelty And Significance:** 3
**Empirical Novelty And Significance:** 3
**Recommendation:** 8
**Confidence:** 5

**Main Review:**


### strengths
- I like the idea of using the relationship of reactants to products for pretraining.
- the paper is well explained
- the experimental results are reasonable (but see comment below)

### weaknesses

The experiments that the authors ran are reasonable to showcase the new method, but are maybe not the most obvious ones I would have expected. The chemical property prediction experiment is good, however, the product ranking experiment (I am a bit reluctant to call it reaction prediction, because no novel product structure is actually predicted), and the graph edit distance prediction are a bit "non-standard".

I would suggest to consider a "proper" reaction prediction task, for example either to use the input representation for template prediction, or have a GNN-to-SMILES-seq transformer model and compare to a SMILES-to-SMILES transformer.

another stronger experiment would be reaction classification, or yield prediction. here, it might be instructive to compare to the work of Schwaller et al. e.g.
https://iopscience.iop.org/article/10.1088/2632-2153/abc81d/meta
https://www.nature.com/articles/s42256-020-00284-w
in this context it also make sense to compare to reaction fingerprints such as the one by Probst and Schneider
https://chemrxiv.org/engage/chemrxiv/article-details/60e358fb379e8d3ba9f92d15
https://pubs.acs.org/doi/abs/10.1021/ci5006614

In my opinion, with these additional experiments the paper would stronger. In the currently form I would only recommend weak accept because I like the concept.



### notes, questions and comments:
1. I found the observation that bond types are not needed in the GNN quite interesting.
2. is the assumption that the reactants and products are sets sufficient, or would a more general assumption that reactants and products are multi-sets (ie. unordered collections where a reactant or product can occur not just once, e.g. in an Ullmann coupling Ph-Br + Br-Ph -> Ph-Ph
) be more expressive?
3. in the derivation it seems that balanced reactions are required, but USPTO is (I believe) not balanced (e.g. leaving groups are disappearing from the right side). would that be a problem?
4. suggested additional references: Merkwirth 2005 for GNNs for molecules https://doi.org/10.1021/ci049613b ; Segler et al 2017  https://doi.org/10.1002/chem.201605499 for reaction prediction with neural nets

**Summary Of The Paper:**

The authors describe a novel pre-training method to learn molecular representation based on the formal relationship between molecules given by chemical reactions (the reactants are related to the products of a chemical reaction)

The performance of the new embedding is evaluated on several chemical tasks.

**Summary Of The Review:**

Creative use of data for pretraining, experiments could be stronger


_________
after author answer:
Score adjusted to 8

---

> ### Author Response · Authors · 2021-11-18
> **Response to Reviewer 17vx**
>
> Thank you so much for your valuable comments. We have carefully revised our paper according to your advice. Our response to your questions are as follows:
>
> ================================================================
>
> Q1: No novel product structure is actually predicted in the product ranking experiment, which makes it a bit “non-standard”.
>
> > It is true that in the experiment of chemical reaction prediction, we are not really “generating” the product but instead ranking the possible candidate products. This is because in this work, we focus on learning molecule representation rather than molecule generation. Therefore, we only design an “encoder” which maps molecules to embeddings, but we do not design a “decoder” that can directly reconstruct a molecule using its embedding. Of course, we can attach a decoder to our model so that our model can do the real product prediction, but this is not the focus of this work.
> >
> > We would like to emphasize that our experimental setting of chemical reaction prediction makes sense, because the number of possible products of a chemical reaction is actually quite limited according to the conservation of atoms and the number of bonds an atom can have. This means that we can enumerate all possible products first then use our model to do the ranking. This experimental setting is also adopted by Jin et al. [1]
> >
> > [1] Jin, Wengong, et al. “Predicting organic reaction outcomes with Weisfeiler-Lehman network.” Proceedings of the 31st International Conference on Neural Information Processing Systems. 2017.
>
> ================================================================
>
> Q2: The experiment of graph edit distance prediction is also a bit "non-standard".
>
> > It is true that the experiment of graph edit distance prediction was first proposed by us, but it is still a standard experiment similar to all other prediction tasks on molecules. The purpose of this task is to see if the learned molecule embeddings can encode molecule structures in terms of their structural similarity. Moreover, this task makes practical sense because calculating the exact GED is an NP-hard problem.
>
> ================================================================
>
> Q3: I would suggest other tasks, for example, to use the input representation for template prediction, to have a GNN-to-SMILES-seq transformer model and compare it to a SMILES-to-SMILES transformer, reaction classification, or yield prediction.
>
> > We followed your advice and added a new experiment of chemical reaction classification. Please refer to Section 3.3 for details.
>
> ================================================================
>
> Q4: Is the assumption that the reactants and products are sets sufficient, or would a more general assumption that reactants and products are multi-sets be more expressive?
>
> > Thanks for the great question. Actually, the assumption that reactants and products are multi-sets is more expressive, because an unbalanced chemical reaction does not actually satisfy the conservation of mass/atom/charge, which is against our goal of preserving such equivalence in the embedding space. For example, for the reaction of CH3CHO + O2 -> CH3COOH, its balanced version is 2CH3CHO + O2 -> 2CH3COOH. It is more reasonable to let 2h(CH3CHO) + h(O2) = 2h(CH3COOH), rather than h(CH3CHO) + h(O2) = h(CH3COOH).
> >
> > The reason why we treat reactants and products as sets rather than multi-sets are two-fold: (1) Organic reactions are usually unbalanced, and they often omit small or inorganic molecules to highlight the product of interest. The USPTO dataset follows this convention as well. Therefore, treating reactants and products as sets is a more realistic assumption, which enable us to make use of these real datasets. (2) Although chemical reactions can be unbalanced under the set assumption, our model can still learn a meaningful reaction template as long as they are written in a consistent manner. For example, as long as all reactions of aldehyde oxidation are written in the form of  R-CHO + O2 -> R-COOH, our model can learn this reaction template and predict that the product of R1-CHO + O2 is R1-COOH.
>
> ================================================================
>
> Q5: In the derivation it seems that balanced reactions are required, but USPTO is not balanced. Would that be a problem?
>
> > Please refer to the answer to Q4. Actually, this is explained in remark 2 on page 4.
>
> ================================================================
>
> Q6: Suggested additional references.
>
> > We have added the related work you mentioned in the introduction.

---

> > ### Comment · Reviewer_17vx · 2021-11-26
> > **accept**
> >
> > Dear authors, thank you for your answers.
> >
> > I think the paper is experimentally solid and the approach creative, so it could be accepted in ICLR

---

### Official Review · Reviewer_h7pt · 2021-11-03

**Correctness:** 4
**Technical Novelty And Significance:** 3
**Empirical Novelty And Significance:** 3
**Recommendation:** 8
**Confidence:** 4

**Main Review:**

Overall, this is an insightful paper.
Although the method is simple (only 0.5 page), it makes sense and works well.
The motivation, the solution and the results are convincing. The paper is clearly written.
In particular, the paper presents several fail cases and tries to explain them.

Please consider and discuss the following:
- In experiments, USPTO which contains 478,612 chemical reactions is used. Did you screen out useless reactions? Or If you directly consider all reactions?
- Report the computation cost.
- Other choices of (5)? Like more complicated pooling methods?
- Discuss the difference and related w.r.t. other papers which consider taking chemical reaction equations as prior knowledge [1,2]

[1] GENERATING MOLECULES VIA CHEMICAL REACTIONS, ICLR 2019 workshop.
[2] A generative model for molecule generation based on chemical reaction trees, ArXiv 2021.


**Summary Of The Paper:**

The paper proposes a molecule representation learning method which is guided by chemical reactions.
In particular, it leverages chemical reaction equations by forcing the sum of reactant embeddings and the sum of product embed- dings to be equal for each chemical equation.
This idea is simple and useful, sharing the spirit of Word2Vec and TransE.





**Summary Of The Review:**

This paper is insightful, clearly written and presents good empirical results.
I vote for acceptance. If the concerns mentioned above are addressed, it can be better.

---

> ### Author Response · Authors · 2021-11-18
> **Response to Reviewer h7pt**
>
> Thank you so much for your valuable comments. We have carefully revised our paper according to your advice. Our response to your questions are as follows:
>
> ================================================================
>
> Q1: For the USPTO dataset, did you screen out useless reactions, or if you directly consider all reactions?
>
> > The original USPTO dataset indeed contains some noisy, incorrect, and repeated chemical reactions. What we used in this paper is not the original version of the USPTO dataset, but the one cleaned by Zheng et al. [1]. We have carefully checked this cleaned version and didn’t see “useless” reactions, so we used the whole dataset in our experiments.
> >
> > [1] Shuangjia Zheng et al. “Predicting retrosynthetic reactions using self-corrected transformer neural networks”. Journal of Chemical Information and Modeling, 60(1):47–55, 2019
>
> ================================================================
>
> Q2: Report the computation cost.
>
> > We added the result of time cost per epoch and maximal memory cost of our model, which is shown in Table 5. Please refer to page 16 for details.
>
> ================================================================
>
> Q3: Other choices of (5)? Like more complicated pooling methods?
>
> > We followed your advice and replaced the SumPooling function in our model with more pooling functions, including AvgPooling, MaxPooling, Attention, SortPooling, Set2Set, and SetTransformer. The results are reported in Table 6. Please refer to the highlighted part on page 16 for details.
>
> ================================================================
>
> Q4: Discuss the difference and relation w.r.t. other papers which consider taking chemical reaction equations as prior knowledge.
>
> >  We added the two related works you mentioned and discussed the difference between these works and ours. Please refer to the highlighted part on page 9 for details.

---

> > ### Comment · Reviewer_h7pt · 2021-11-29
> > **Thanks for the reply.**
> >
> > Dear authors, thanks for the reply and revision in the paper. My concerns are cleared. Overall, I think this paper is insightful, interesting and simple to use, which has high potential to influence the society. Thus, I raise my score.

---

### Decision · Program_Chairs · 2022-01-20

**Decision:**

Accept (Poster)

**Comment:**

This paper uses chemical reaction data as a means to help train molecule embeddings, by requiring embeddings to satisfy known reaction equations. The idea is nice and clear, and the paper includes strong empirical evaluation. All four reviewers agreed the paper could be accepted, with two of them raising their scores after a detailed author rebuttal and discussion, which included additional experiments.